# Preclinical Experience of the Mayo Spheroid Reservoir Bioartificial Liver (SRBAL) in Management of Acute Liver Failure

**Philipp Felgendreff** [1,2,†], **Mohammad Tharwat** [1,3,†], **Seyed M. Hosseiniasl** [1,†], **Bruce P. Amiot** [1], **Anna Minshew** [1], **Anan A. Abu Rmilah** [1], **Xiaoye Sun** [1], **Dustin Duffy** [1], **Walter K. Kremers** [1] and **Scott L. Nyberg** [1,4,*]

1 Department of Surgery, Mayo Clinic, Rochester, MN 55902, USA
2 Department for General, Visceral and Vascular Surgery, University Hospital Jena, 07743 Jena, Germany
3 General Surgery Department, Faculty of Medicine, Zagazig University, Zagazig 44519, Egypt
4 William J. von Liebig Center for Transplantation and Clinical Regeneration, Mayo Clinic, Rochester, MN 55902, USA
* Correspondence: nyberg.scott@mayo.edu
† These authors contributed equally to this work.

**Abstract:** The Spheroid Reservoir Bioartificial Liver (SRBAL) is an innovative treatment option for acute liver failure (ALF). This extracorporeal support device, which provides detoxification and other liver functions using high-density culture of porcine hepatocyte spheroids, has been reported in three randomized large animal studies. A meta-analysis of these three preclinical studies was performed to establish efficacy of SRBAL treatment in terms of survival benefit and neuroprotective effect. The studies included two hepatotoxic drug models of ALF (D-galactosamine, α-amanitin/lipopolysaccharide) or a liver resection model (85% hepatectomy) in pigs or monkeys. The SRBAL treatment was started in three different settings starting at 12 h, 24 h or 48 h after induction of ALF; comparisons were made with two similar control groups in each model. SRBAL therapy was associated with significant survival and neuroprotective benefits in all three animal models of ALF. The benefits of therapy were dose dependent with the most effective configuration of SRBAL being continuous treatment of 24 h duration and dose of 200 g of porcine hepatic spheroids. Future clinical testing of SRBAL in patients with ALF appears warranted.

**Keywords:** Spheroid Reservoir Bioartificial Liver (SRBAL); acute liver failure (ALF); bioartificial liver (BAL); liver support devices



## 1. Introduction

Acute liver failure (ALF) results from severe hepatocellular injury of a previously healthy liver and causing significant impairment of anabolic, catabolic and detoxification function associated with impaired mental status (hepatic encephalopathy) [1]. Furthermore, the liver injury is associated with coagulopathy, impaired detoxification, elevated levels of serum ammonia and systemic inflammatory effects in the patients. In particular, the extrahepatic manifestations of systemic inflammation and elevated serum ammonia include renal dysfunction, pulmonary dysfunction, impairment of the blood–brain barrier, astrocyte edema and brain swelling [2,3], Progression of these events leads to intracranial hypertension, cerebral herniation, and brain death as a fatal consequence of ALF [2,4]. The overall mortality rate of ALF patients range from 30% to 75% [5]. The mortality rate of ALF varies depending on its etiology [6]. Currently, orthotopic liver transplantation remains the only definitive treatment option for ALF patients [7]. Due to the limited resources as well as the global organ shortage, world-wide fewer than 10% of ALF patients receive this life saving therapy [8,9].

One promising alternative therapeutic option to liver transplantation is an extracorporeal liver supporting system to support the acutely injured liver during its regeneration

and recovery. Since 1955 [10], numerous extracorporeal liver support or "liver dialysis" approaches have been pursued intensively in both preclinical and clinical studies [11]. Categorization of these approaches into purely artificial and bioartificial liver support devices has emerged [12].

The artificial liver devices use physical or chemical gradients and adsorption to eliminate toxins and metabolic waste in the blood or plasma of the patients [13]. In clinical practice, four central artificial systems are currently in use: molecular adsorbent recirculating system (MARS), single-pass albumin, dialysis (SPAD), fractionated plasma separation and adsorption system (Prometheus), and selective plasma filtration therapy (SEPET) [14]. Clinical studies demonstrated the effect of these devices by reducing serum bilirubin, serum creatinine and urea during the treatment period in ALF patients [15,16]. However, the treatment-associated improvement in singular parameter has not been associated with a survival benefit without the possibility of a lifesaving liver transplant. Artificial support devices may serve as a bridge to transplant for some patients with ALF, acute-on-chronic liver failure, hepatorenal syndrome, severe cholestasis or hepatic encephalophy [17,18]. However, to date, no artificial liver support device has been shown to fully replace liver function nor has such a device improved patient survival alone without liver transplantation.

The bioartificial liver support devices (BAL) were first reported by Matsumura et al. [19] and Margulis et al. [20] over two decades ago. These cell-based devices provide complex detoxification of metabolic wastes, elimination of waste materials, and synthetic biochemical processes of living cells. The extracorporeal BAL devices are designed are designed with semipermeable membranes enabling either direct or indirect contact with the patient's blood stream and the required metabolic active liver cells. The membranes may serve as an immunoprotective barrier to the cell-based device [15].

BAL devices have utilized a variety of cell types including immortalized human cell lines such as the human hepatoblastoma lines HepG2 and C3A [21], human hepatocytes [22], and porcine hepatocytes [23,24]. The immortalized lines were less difficult to expand in vitro [25], and produced human proteins such as human albumin, however they showed incomplete expression of metabolic enzymes (like urea cycle enzymes [26,27]). Consequently, primary hepatocytes are preferred for BAL by some investigators because of their complete repertoire of liver functions. Human hepatocytes would be ideal for BAL application, but their quantities and availability are currently not practical for clinical usage. Therefore, it is our opinion that porcine hepatocytes are the preferred cell source for BAL application currently. Porcine hepatocytes have a similar metabolic profile to human hepatocytes and are available in large quantities.

Regardless of the source of hepatocytes, the traditional culture of primary hepatocytes is problematic and has been associated with limited inoculation, rapid loss of differential function, and premature cell death [19,20,28]. Establishing primary 3-dimension spheroid culture of hepatocytes by a rocked suspension method has been shown to circumvent the challenges of traditional monolayer culture of hepatocytes. During rocked suspension culture, individual hepatocytes adhere and form stable spheroids spontaneously by linking to each other via membrane-bound E-cadherin [29]. The analysis of these spherical aggregate cultures shows a regular gene expression profile, normal Cyp-isoenzymes, and an intact urea cycle for ammonia detoxification. Next to maintaining the functionality of the spheroids for at least 24 h up to weeks [30], these spheroids can be easily scaled-up (1 g of hepatic spheroids contains $1 \times 10^8$ hepatocytes) and be used in higher concentration to be suitable for BAL devices [31].

A cell-based liver support device containing such hepatocyte spheroids is the Mayo Spheroid Reservoir Bioartificial Liver (SRBAL). The SRBAL device encompasses two independent extracorporeal perfusion circles connected via a hollow fiber cartridge enabling an exchange of molecules between the patient's blood and the contained spheroids (Figure 1). In this setup, the first circle is containing blood of the patient and the second circle is loaded with human albumin primed solution (5 g/L) connected to the reservoir containing the spheroids (Video S1). Using both convection and diffusion waste, and toxin molecules

in the patient's blood pass through the hollow fibers into the albumin solution of the second circuit. The waste and toxin molecules (especially ammonia) are metabolized in the reservoir by the contained spheroids to urea as well as other secondary metabolic products. Following elimination of the urea, the purified albumin solution is available again as an exchange medium, ensuring continuous recirculation of the hollow fiber cartridges. This system allows a continuous treatment of the patients, by avoiding the direct contact of the patient blood and the extracorporeal hepatocytes. In this way, allogeneic or xenogenic treatment constellations can be safely realized. The treatment efficacy of this system was already confirmed in bench studies [31]. Furthermore, the SRBAL device was used in established ALF large animal studies showing the success of SRBAL treatment individually.

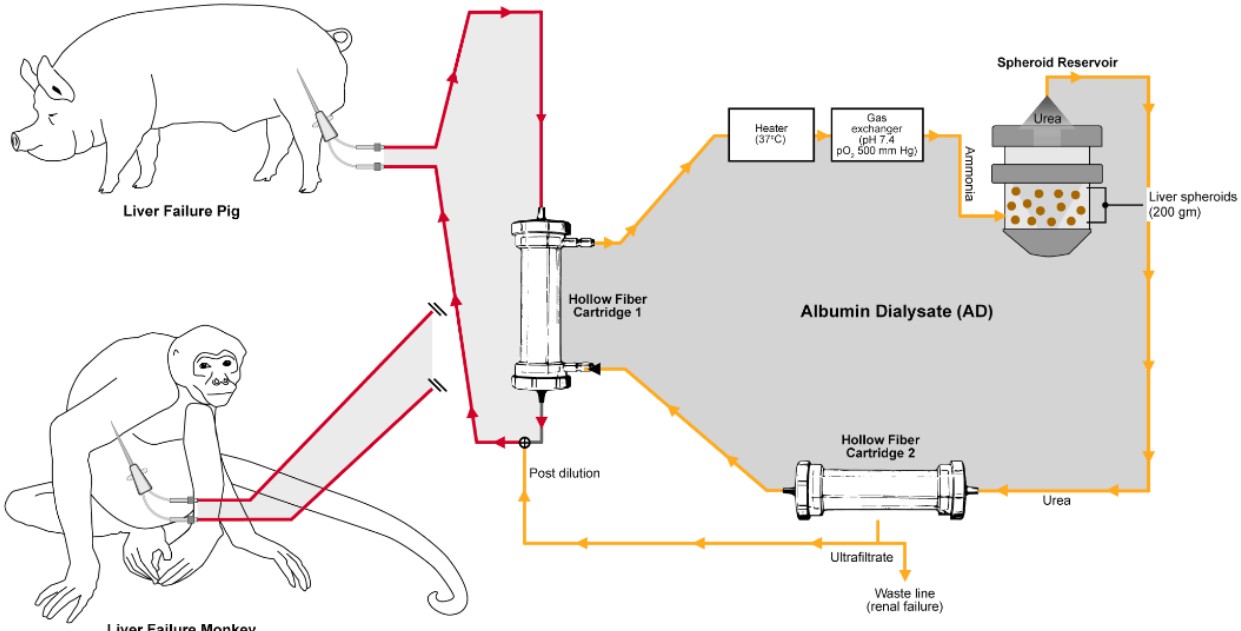

**Figure 1.** Illustration of the Mayo Spheroid Reservoir Bioartificial Liver (SRBAL) device perfusion circle including the Spheroid reservoir, the hollow fiber cartridges, and the blood (red) and albumin (yellow) circle.

Currently an overview of bench and preclinical applications of SRBAL is missing. Therefore, the aim of this review and meta-analysis is both an overview and the comparison of SRBAL treatment in three independent large animal ALF models. Of particular interest is the effect of the SRBAL device on the replacement of liver function in these clinically relevant large animal models. Furthermore, impact of therapy on parameters like ammonia level, cytokine expression and liver specific serum parameter as well as survival duration will be focuses of this report. This analysis of preclinical data will be used in establishing efficacy of SRBAL therapy and warrant its first clinical application.

## 2. Methods

The results of three independent preclinical studies of SRBAL treatment were combined and analyzed collectively. In each case, an established large animal models of ALF was utilized (Glorioso et al., 2015 [32], Li et al., 2018 [33] and Chen et al., 2019 [29]). Porcine hepatocytes were isolated by 2-step perfusion method and hepatocyte spheroids were formed by rocker technique as previously reported [31]. The reservoir compartment of SRBAL, shown in Figure 1, was filled with fresh hepatocyte spheroids prior to each treatment.

All studies compared the effects of SRBAL treatment vs. standard medical therapy alone (SMT) or SMT plus SRBAL with no cells in the reservoir (NCBAL). SMT was used in all experiments and a similar SRBAL extracorporeal circuit (with or without hepatocyte

spheroids) was used to avoid study bias. All treatments in each individual study were performed under mild sedation by continuous intravenous administration of propofol (0.1–0.2 mg/kg/min). The treatment effect of SRBAL in all study arms was determinate by laboratory results (ammonia, bilirubin, AST blood levels, coagulation profile), ICP measurements and cytokine levels (TNF-$\alpha$ and IL-6) with resulting impact on the represented survival rate and duration. Other study variables investigated porosity of SRBAL membrane, dose of hepatocyte spheroids per treatment, time from ALF induction to initiation of SRBAL treatment, and duration and frequency of SRBAL treatments, as summarized in Figure 2.

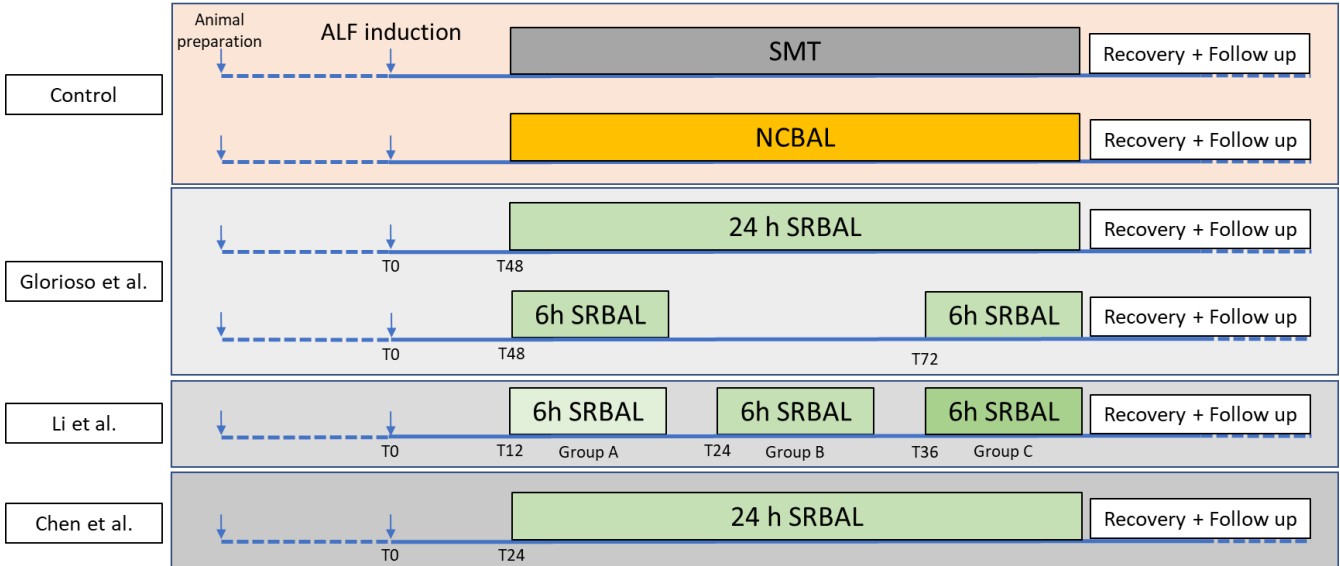

**Figure 2.** Design of all studies including the timepoint of ALF induction, the start and duration of the SRBAL treatment and the reprehensive control groups (SMT and NCBAL) [29,32,33].

### 2.1. Study #1—Pivotal Preclinical Trial of the Spheroid Reservoir Bioartificial Liver—Glorioso et al. [32]

The pivotal preclinical study of SRBAL treatment by Glorioso et al. established SRBAL treatment in a drug induced model of ALF [32]. For this study, ALF was induced in domestic pigs (range 43.8–51.0 kg) by toxic dose of 0.75 g/kg BW D-galactosamine. Forty-eight hours after ALF induction, six animals per group were treated by SMT alone, SMT + NCBAL or SMT + SRBAL. Spheroid dose of SRBAL ranged from 59–228 g porcine hepatocytes. SRBAL device used membranes with either 70 kD or 400 kD porosity. To investigate the impact of treatment duration, animals receiving extracorporeal treatment were divided equally between continuous 24 h treatment or intermittent treatment (2 × 6 h treatment rounds with 18 h non-treatment interval). All animals were observed for up to 90 h after ALF induction.

### 2.2. Novel Spheroid Reservoir Bioartificial Liver Improves Survival of Nonhuman Primates in a Toxin-Induced Model of Acute Liver Failure—Li et al. [33]

The second preclinical study of SRBAL device was published by Li et al. in 2018 [33]. This follow-up study investigated the treatment effect in a nonhuman primate model of ALF. Rhesus monkeys received the hepatotoxic dose of 0.1 mg/kg $\alpha$-amanitin and 1.0 µg/kg lipopolysaccharide to induce ALF. The 6 h SRBAL treatment period started 12, 24 or 36 h after ALF induction and was compared to SMT alone, or no cell extracorporeal therapy. In addition to standard laboratory parameters, cytokines associated with the inflammatory response (TNF-$\alpha$, IL-6, IL-12, IL-1$\beta$, IL-8, IFN-$\gamma$, IL-12, and anti-inflammatory cytokine IL-10), tissue proliferation (HGF, EGF, VEGF) and hematopoietic proliferation (M-CSF including IL-1RA, macrophage migration inhibitory factor (MIF)) were measured in monkey serum by ELISA technique. Porcine hepatocytes and peripheral blood mononuclear

cells of treatment animals were screened by ddPCR and qPCR for presence of porcine endogenous retrovirus (PERV), an important marker of xenozoonosis. The functionality of the blood–brain barrier was evaluated by measuring levels of S-100 β protein in peripheral blood. Study monkeys were monitored closely during study interval of 14 days after ALF induction, and an additional 1 year to assess long term effects of SRBAL therapy.

*2.3. Randomized Trial of Spheroid Reservoir Bioartificial Liver in Porcine Model of Post Hepatectomy Liver Failure—Chen et al. [29]*

In the third study, Chen et al. [29] investigated the regenerative effects of SRBAL treatment in a post-liver resection porcine model of ALF. Animals underwent 85% liver resection using crash clamp technique according to Court et al. [34]. Anatomic landmarks of the left, median and right lateral liver lobes were followed to limit blood loss (<300 mL) safely and avoid surgical bias. The extent of liver resection was confirmed by CT-volumetry comparing pre-operative liver volume to remnant volume immediately after surgery. Twenty-four hours after liver resection, animals were randomized in the three experimental groups (SMT, SMT + NCBAL and SMT + SRBAL) of 24 h continuous duration. SRBAL treatments utilized 200 g of porcine hepatocyte spheroids. To investigate the pro-regenerative effect of treatment, additional CT scans were obtained and remnant liver volume of study animals was determined 43 h and 90 h after liver resection. Histological (H&E staining) and Ki-67-immunohistochemical stains were also used to assess liver regeneration. In addition to standard liver function tests, serum markers of systemic inflammation (tumor necrosis factor-α (TNF-α), transforming growth factor-β (TGF-β) and Interleukin-6 levels) were measured by ELISA technique. Animals were monitored closely until the primary study endpoint of 90 h after liver resection; survival was also assessed at 2 weeks after liver resection.

## 3. Statistics

Statistical analysis was performed using R v4.0.3 (R Foundation for Statistical Computing, Vienna, Austria). All continuous variables of each study were reported as median (Mdn) and range (R) or as mean (M) and standard deviation (±SD). Mean differences in biological variables between groups were analyzed by using analysis of variance (ANOVA). Differences in the incidence of time-to-event outcomes were tested using the log-rank test. For the meta-analysis of the three included papers, the previously reported $p$-values were assessed using Fisher's method. A $p$-value less than 0.05 was considered significant.

## 4. Results
### 4.1. Overview

The SRBAL treatment was investigated in three individual studies using ALF large animal models. These studies included a total of 66 animals (36 domestic pigs and 30 rhesus monkeys), with 45.45% male and 54.55% female animals (Table 1). The average body weight of the pigs was 45 kg (±1.7) (Glorioso et al.) and 28 kg (±1.2) (Chen et al.), respectively. The rhesus monkey had an average body weight of 7.7 kg (±0.4) and were 5–7 years old. The analysis of baseline animal characteristics and laboratory values showed no significant differences between experimental groups. In two studies the ALF was drug induced (48 animals) by applying D-galactosamine (18 domestic pigs) or α-amanitin and lipopolysaccharide (30 rhesus monkeys). Chen et al. induced ALF by 85% hepatectomy (18 domestic pigs). SRBAL treatments were conducted for a range of durations, up to 24 h, and initiated at standardized time points after induction of ALF based on characteristics of the ALF model and different objectives of each study.

The viability of hepatocytes after isolation ranged from 96.2% and 98%. The inoculated mass of hepatocyte spheroids used in all studies ranged between 59 g and 228 g (average: 119 g).

**Table 1.** Summarizing the general experimental data of all included SRBAL preclinical treatment studies.

| Author and Year | Glorioso et al. [32] | Li et al. [33] | Chen et al. [29] |
|---|---|---|---|
| Species | Domestic pig | Rhesus monkey | Domestic pig |
| Animal number [n] | 18 | 30 | 18 |
| Sex [F/M] | F | M | F |
| Average body weight [kg] | 45.0 (±1.7) | 7.7 (±0.4) | 28.0 (±1.2) |
| ALF induction method | Drug induced ALF | Drug induced ALF | surgical induced ALF |
| Concentration of applied drug | 0.75 g/kg D-galactosamine | 0.1 mg/kg α-amanitin + 1.0 µg/kg lipopolysaccharide | - |
| Resected liver lobs | - | - | Left lateral, medial right lateral liver lobe |
| Start of individual treatment after ALF induction | T48 | T12, T24, T36 | T24 |
| Duration of treatment [hours] | 2 × 6 h or 24 h | 6 h | 24 h |
| Average spheroid mass [g] | 117.4 (±48.3) | 100.2 (±3.3) | 207.9 (±21.8) |
| Hepatocytes viability [%] | 96.7 (±2.2) | 98.0 (±1.0) | 96.2 (±1.9) |
| Source of hepatocytes | Domestic pig | Bama miniature pig | Domestic pig |

### 4.2. Survival

The SRBAL treatment led to significant survival benefit in all three ALF models (Figure 3). Independent of ALF model (drug or surgical) or the species (domestic pig or rhesus monkey), the SRBAL led to a survival improvement compared to SMT alone (72 h: $p < 0.001$; 90 h: $p < 0.001$) or SMT + NCBAL (72 h: $p < 0.001$; 90 h: $p < 0.001$) study groups (Table 2). In addition, Li et al. and Chen et al. were able to demonstrate a significant improvement in survival in the SRBAL group compared to the SMT alone and SMT + NCBAL groups in the long-term analysis.

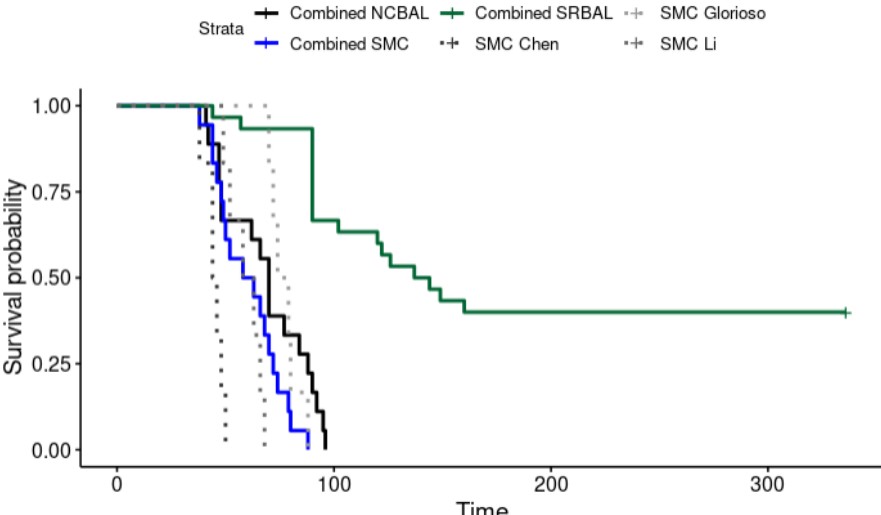

**Figure 3.** Kaplan–Meier survival curve of all three investigated treatment options (SMT: standard medical therapy, NCBAL: SRBAL with no cells in the reservoir, SRBAL: Mayo Spheroid Reservoir Bioartificial Liver) in comparison to the expected survival curve of each individual model. Each curve represents the cumulative survival rate of all three included original article (Glorioso et al. [32] Li et al. [33] Chen at al. [29]) in consideration of the individual treatment regime and the expected survival rate of each individual ALF model.

**Table 2.** Survival *p* value in SRBAL groups vs. SMT and NCBAL groups in each study as well as the combined *p* values of all studies at the following time points: 72, 90, 336 h.

| Comparison | Study | 72 h Survival | 90 h Survival | 336 h Survival |
|---|---|---|---|---|
| SMT vs. SRBAL | Glorioso et al. | 0.9 | 0.01 | N/A |
| | Li et al. | 0.001 | 0.001 | 0.001 |
| | Chen et al. | <0.001 | <0.001 | <0.001 |
| | Combined value | <0.001 | <0.001 | <0.001 |
| NCBAL vs. SRBAL | Glorioso et al. | 0.1 | 0.05 | N/A |
| | Li et al. | 0.080 | 0.001 | 0.001 |
| | Chen et al. | 0.001 | 0.001 | 0.001 |
| | Combined value | <0.001 | <0.001 | <0.001 |

Furthermore, Li et al. showed a significant increase in the survival rate by initiating the SRBAL treatment shortly after ALF induction. Rhesus monkeys, who underwent SRBAL treatment 12 h after ALF induction, survived for an average of 336 h. When treatment was started later, 36 h after induction, the average survival of treated animals was reduced to 131.5 h.

Glorioso et al. and Chen et al. confirmed a survival benefit of SRBAL therapy at 90 h in both drug-induced and surgery-induced models of porcine ALF, respectively. Five of six pigs in Chen's study survived to the primary endpoint (90 h); one animal experienced brain herniation 44 h after ALF induction. Three of the five animals in Chen's study that were alive at 90 h were also alive after 14 days; the other two animals were euthanized during the second week after 85% liver resection when they experienced small-for-liver size symptoms of tense ascites and poor oral intake. These studies also concluded than the amount of liver spheroids contained in SRBAL as well as the continuous treatment duration of 24 h were important conditions ensuring survival after ALF induction [29,32].

*4.3. Serum Ammonia Levels and Other Liver Specific Serum Parameters*

The analysis of all included studies showed SRBAL devices were effective in decreasing serum ammonia levels and improving liver-specific parameters in all ALF models. In case of the serum ammonia levels, the SRBAL treatment was leading to a significant reduction of the ammonia levels during both, the treatment and entire study period in comparison to the SMT and the NCBAL in all three studies (Table 3).

Additionally, early initiation of treatment resulted in the most significant reduction in serum ammonia levels at 48 h after ALF induction. Li et al. showed a decrease of serum ammonia from $740.0 \pm 21.2$ μM in the NCBAL group to $114.8 \pm 9.6$ μM in the SRBAL group, if the treatment started 12 h after administration of hepatotoxin. Furthermore, the Glorioso study reported a marked deceleration in the rise of serum ammonia during SRBAL treatment ($+5.1 \pm 15.0$ μg/dL/hr) while serum ammonia increased sharply in the control groups (NCBAL group: $+20.8 \pm 13.1$ μg/dL/h; SMT: $+23.6 \pm 11.1$ μg/dL/h) during the same interval, 48 h to 72 h after ALF induction. The difference in ammonia levels between treatment and control groups can be explained by direct or indirect detoxification effects of SRBAL treatment. The direct effect is supported by ammonia detoxification measured in the SRBAL reservoir; the indirect effect is supported by increased regeneration of the liver remnant in association with SRBAL treatment. These findings are supported by liver-specific serum markers such as AST and Bili measured 72 h and 90 h after ALF induction. Specifically, significant differences in AST and Bili were detected in comparison of the SRBAL vs. SMT vs. NCBAL groups as shown in Table 3.

**Table 3.** Comparison of AST, bilirubin (Bil), ammonia levels (NH3) of SRBAL treatment vs. SMT and NCBAL groups in each study along with corresponding *p*-values.

| Time Point | Serum Parameter | SMT vs. SRBAL | | | | BAL vs. SRBAL | | | |
|---|---|---|---|---|---|---|---|---|---|
| | | Glorioso et al. [32] | Li et al. [33] | Chen et al. [29] | Fischer's Method | Glorioso et al. [32] | Li et al. [33] | Chen et al. [29] | Fischer's Method |
| 0 h | AST | 1.000 | 0.512 | 0.761 | 0.930 | 1.000 | 0.722 | 0.937 | 0.993 |
| | Bil | 1.000 | 0.341 | 0.341 | 0.636 | 1.000 | 0.341 | 0.341 | 0.636 |
| | NH3 | 1.000 | 0.182 | 0.732 | 0.673 | 1.000 | 0.940 | 0.508 | 0.961 |
| 12 h | AST | 1.000 | 0.369 | 0.210 | 0.529 | 1.000 | 0.064 | 0.946 | 0.468 |
| | Bil | 1.000 | 0.640 | 0.475 | 0.881 | 1.000 | 0.934 | 0.783 | 0.996 |
| | NH3 | 1.000 | 0.134 | 0.230 | 0.325 | 1.000 | 0.136 | 0.174 | 0.278 |
| 36 h | AST | 0.006 | 0.876 | 0.563 | 0.072 | 0.466 | 0.062 | 0.508 | 0.208 |
| | Bil | 0.005 | 0.716 | 0.760 | 0.062 | 0.016 | 0.888 | 0.438 | 0.118 |
| | NH3 | 0.022 | 0.002 | 0.702 | 0.002 | 0.603 | 0.019 | 0.494 | 0.111 |
| 48 h | AST | 0.137 | 0.343 | 0.902 | 0.387 | 0.663 | 0.052 | 0.767 | 0.296 |
| | Bil | <0.001 | 0.837 | 0.728 | 0.002 | <0.001 | 0.551 | 0.761 | 0.008 |
| | NH3 | 0.003 | <0.001 | 0.366 | <0.001 | 0.026 | 0.013 | 0.636 | 0.010 |
| 72 h | AST | 0.225 | 0.022 | 0.124 | 0.022 | 0.957 | 0.007 | 0.446 | 0.070 |
| | Bil | 0.001 | 0.939 | 0.202 | 0.006 | 0.001 | 0.453 | 0.342 | 0.006 |
| | NH3 | 0.004 | <0.001 | 0.055 | <0.001 | 0.002 | 0.004 | 0.066 | <0.001 |
| 90 h | AST | 0.512 | 0.013 | 0.179 | 0.037 | 0.691 | 0.005 | 0.567 | 0.049 |
| | Bil | 0.007 | 0.812 | 0.326 | 0.052 | 0.008 | 0.658 | 0.722 | 0.082 |
| | NH3 | 0.007 | 0.000 | 0.141 | <0.001 | 0.005 | 0.004 | 0.193 | 0.000 |

*4.4. Neuroprotection of SRBAL Treatment in Animal Models of ALF*

Brain integrity and brain function was protected during SRBAL treatment compared to control groups in all three studies. Animals in SMT alone and NCBAL + SMT groups were associated with elevated markers of brain integrity (ICP or S-100 β levels) during the study period. These neurological abnormalities were independent of animal species or etiology of ALF. Both of these direct and indirect markers of brain integrity correlated with the rise in blood ammonia and increased risk of death from ALF. In contrast, ICP levels were significantly lower at the end of SRBAL treatment compared with the SMT alone and NCBAL + SMT. Continuous 24 h SRBAL treatment was associated with the lowest ICP levels and greatest level of neuroprotection.

Similarly, serum levels of S-100 β were decreased by SRBAL treatment compared to control groups, and this benefit was greatest if SRBAL therapy was initiated earlier. S-100 β serum levels 48 h after induction of the ALF: SMT-group: 27.1 ± 5.0 ng/mL; NCBAL-group: 16.4 ± 6.0 ng/mL; SRBAL-treatment started 6 h after ALF induction: 5.5 ± 1.6 ng/mL; SRBAL-treatment started 36 h after ALF induction: 14.3 ± 1.2 ng/mL). These results underlined the protective aspect of the SRBAL device in terms of the brain integrity in all included studies.

*4.5. Effect of SRBAL on Proinflammatory Cytokines and Transmission of Xenozoonosis*

The effect of SRBAL treatment of the serum levels in proinflammatory cytokines was indifferent and did not correlate with the survival or outcome of the treatment. Especially, Glorioso et al. detected a wide range of IL1β, IL6, IL18 and TNFα without any effect on the study related endpoint parameters. Similar results were reported by Chen et al. IL-6

and TNFα were affected by the hepatectomy itself but did not correlate with the applied treatment regime.

In addition, a transmission of xenozoonosis during the SRBAL treatment of xenogenic AFL were ruled out in the study of Li et al. By using both ddPCR and qPCR techniques, no evidence of PERV was found in porcine hepatocytes spheroids or in peripheral blood mononuclear cells from rhesus monkeys. This supported the principle xenogeneic utilization of SRBAL treatment, even if this is associated with highly legal and ethical hurdles.

## 5. Discussion

The beneficial effects of SRBAL therapy in preclinical models of ALF have been previously reported individually. This meta-analysis underscores these benefits collectively in terms of survival duration, improvement in liver function, neuroprotection, and severity/development of hepatic encephalopathy. The present study is therefore a valuable complement to the individual findings of the included more comprehensive studies. For example, in addition to the general benefits of SRBAL treatment, two of the three studies [29,32] also identified limited improvement on individual laboratory parameters such as INR. However, these individual findings were not included in the present analysis to avoid a bias in the comparison of the three ALF treatment options.

Overall, across three different large animal recovery models of ALF, SRBAL treatment improved survival compared to the SMT alone and NCBAL therapy, a control used because of its similarity and clinical relevance to albumin dialysis.

Furthermore, this meta-analysis tested for the effect of different treatment durations, different timepoints of initialing the treatment and different amounts of hepatic spheroids in the SRBAL device.

Due to the liver specific dynamic of ALF [2,3], an early initiation of SRBAL treatment as continuous 24 h therapy using a dose of 200 g hepatocyte spheroids showed the highest efficiency in treatment of ALF. The success of SRBAL treatment was demonstrated in robust preclinical studies—regardless of the used ALF models or the investigated animal species. Based on these results, the SRBAL device can be a innovative therapy option for patients with ALF and potentially evolve as an alternative to the organ transplantation.

A limitation of this analysis may be the choice of D-galactosamine rather than acetaminophen as hepatotoxic drug. Acetaminophen overdose is a leading causes of drug induced ALF in humans [35], but the study of acetaminophen hepatotoxicity in large animals is problematic. Along with liver injury, bolus administration of acetaminophen in pigs and other large animals is associated hemolysis, methemoglobinemia, pulmonary failure and renal failure at doses associated with mild-to-moderate liver injury [36,37]. These side effects can be mitigated with intravenous titration of acetaminophen; however this method is labor intensive and difficult to reproduce, especially in the setting of therapeutic intervention. In contrast, D-galactosamine can be administered as a bolus and its toxicity is liver-specific and reproduceable in most animal models, especially pigs. Elevated transaminases, severe coagulopathy, signs of hepatic encephalopathy and elevated ICP have been observed in pigs [38–40], rabbits [41], dogs [42] and rodents [43–45]. The mechanism of D-galactosamine toxicity is based on its effect on intracellular stores of uridine as well as blockage of the glycogen synthesis [46]. These intracellular pathways of injury are not shared by acetaminophen toxicity. However, the excess dosing of both drugs share histopathological changes on liver biopsy that are felt to be clinically relevant [47] for use of D-galactosamine as a preclinical ALF model.

As expected, surgical resection models of ALF are more comparable in large animals and humans [48]. Commonly used surgical approaches for liver regeneration studies as well as for testing liver supporting devices have included anhepatic, partial hepatectomy and combined ischemia and parenchyma resection procedures. In pigs, Pagano et al. [49] described signs of ALF following a 80% liver resection or more. This extent of resection is surgically challenging and can impact reproducibility of the model. Thus, modifications to liver resection have been proposed such as ischemia preconditioning [50,51] and common

bile duct ligation [52]. The 85% liver resection model was used because of its reproducibility and high rate of reaching a death equivalent endpoint before 72 h in control animals. Clinical relevance to post-resection ALF and small-for-size graft syndrome [47] were other favorable considerations for this model.

Porcine hepatocytes provide an abundant, readily available, not prohibitively expensive cell source for a bioartificial liver. As already mentioned, porcine hepatocytes have a similar metabolic profile to human hepatocytes. In the spheroid configuration the cells maintain stable functionality for days to weeks [30] and can be used in higher concentration to be suitable for BAL devices [31].

Transmission of immunological and infectious agents was another consideration in selecting recipient models for evaluation of a porcine hepatocyte SRBAL. Xeno-zoonosis is a possible risk that must be mitigated during human exposure to porcine hepatocytes [53,54]. This risk was considered and investigated in the study design by Li at al. Of importance, the authors did not detect any evidence of PERV transmission to monkeys up to 1 year after SRBAL treatment. This observation may be explained by the 65 KDa (<5 nm pose size) semipermeable membrane used to prevent transmission of PERV (>25 nm) and other small potentially infectious organisms during SRBAL treatment [55]. This membrane is also immunoprotective as it prevents transfer of the patient's immune components (i.e., immunoglobulins, complement proteins, and white blood cells) into the hepatocyte reservoir while allowing robust transfer of metabolic waste products [56]. Germ-free animal herds [57], experimental primary human hepatocytes expansion approaches like embryo complementation technique [58] or human-pig hybrid cells development [59,60] are also being considered to further reduce the risk of xenozoonosis transmission during xenotransplant procedures. Novel methods to expand primary human hepatocytes remain a holy grail to a humanized SRBAL device.

The study of Glorioso et al. provided preliminary evidence that a minimum mass of liver tissue was required to provide adequate bioartificial liver support during ALF. This mass was 200 g of viable hepatocytes representing $20 \times 10^9$ cells to support a 30 kg pig during recovery from ALF. This amount is not surprising and in fact similar to the 20% of the healthy liver mass predicted to support liver function in humans after liver resection [61,62].

While the results of this meta-analysis offer promising prospects for potential clinical application, prior trials with earlier generations of artificial and bio-artificial liver support devices have shown that translating preclinical results to human ALF patients is challenging. The complexity of ALF and its associated extrahepatic manifestations such as ARDS, MSOF and hepatorenal failure [5], along with the patient's pre-existing comorbidities pose a great challenge. Previous clinical trials have identified the heterogeneous population(s) of ALF patients and the insufficient biological activity of previous liver support devices as primary reasons why these earlier devices have not proven successful in randomized control trials nor gained FDA approval for treatment of ALF patients [24,63,64]. Further research may be useful to identify the optimal selection criteria for ALF patients in clinical trials.

## 6. Conclusions

By combining data from multiple drug and surgical models of ALF, the collective effect of SRBAL therapy in the preclinical setting has been underscored. The SRBAL devices represent a new promising treatment option for patients with ALF. Extensive preclinical testing in three independent large animal models of ALF showed a positive effect on survival and was associated with robust detoxification by the SRBAL device and an associated neuroprotection. Furthermore, safety of the SRBAL device was established in a xenogeneic treatment regime by using a semipermeable membrane to separate the patient's blood from the spheroid reservoir filled with porcine hepatocytes. The SRBAL configuration allows continuous supportive therapy for up to 24 h. Based on these favorable and reproducible results, further evaluation of SRBAL treatment in clinical trials of ALF appears warranted.

**Supplementary Materials:** The following supporting information can be downloaded at: https://www.mdpi.com/article/10.3390/livers2040029/s1, Video S1: Glorioso et al.: Spheroid Reservoir Bio-Artificial Liver [32].

**Author Contributions:** P.F., M.T. and S.M.H. collected data, assistant for formal analysis, writing the original draft and preparing the figures; B.P.A., A.M., A.A.A.R. and X.S. provided study data and validated the statistics results; D.D. and W.K.K. analyzed the formal data, assisted in the methodology and preparing the figures; S.L.N. supervise the analysis, review and editing the manuscript. All authors have read and agreed to the published version of the manuscript.

**Funding:** This research was funded by grants from Deutsche Forschungsgesellschaft (DFG) grant number: FE 2089/1-1 (PF), Egyptian Cultural and Educational Bureau JS 3903 (MT), Mayo Foundation, and NIH (P30DK084567, R01DK106667).

**Institutional Review Board Statement:** Not applicable.

**Informed Consent Statement:** Not applicable.

**Data Availability Statement:** Not applicable.

**Conflicts of Interest:** B.P.A. and S.L.N. are inventors of Mayo SRBAL device. Mayo Foundation has licensed this patent to Ponte Biomedical, Inc. The authors declare no other conflict of interest.

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
