# Peer review of "Preclinical Experience of the Mayo Spheroid Reservoir Bioartificial Liver (SRBAL) in Management of Acute Liver Failure"

_livers, doi:10.3390/livers2040029_

Round 1
Reviewer 1 Report
Artificial liver support continues to be of high interest given the shortage of donor livers. Some limitations of the studies could be discussed more in detail. For example
1) This analysis suggests some short term benefits in acute liver failure and perhaps a longer survival benefit in a hepatectomy model. This is not surprising as residual hepatocytes that are not injured are more likely to regenerate more effectively than those that are damaged from hepatotoxins. Appropriate citations should be included in this aspect.
2) There is a lack of histologic evidence of regeneration of the native liver in the ALF models. This is important since drug induced ALF is a leading cause of liver toxicity and the short term benefits from metabolic support from SBARS does not necessarily address the real world problem of donor shortage unless the metabolic support can be shown to continue until the time that the native liver begins to regenerate to obviate the need to OLT.
3) Benefits of SBARS appears to be maximum when initiated immediately after the induction of ALF and in one of the models, survival decreased significantly when started 12 hours after ALF. This should also be discussed as a limiting factor as patients do not always present to hospitals early in the course of ALF.
Author Response
Dear Reviewer,
thank you very much for reviewing our manuscript. See bellow the answers to your questions.
Q1) This analysis suggests some short term benefits in acute liver failure and perhaps a longer survival benefit in a hepatectomy model. This is not surprising as residual hepatocytes that are not injured are more likely to regenerate more effectively than those that are damaged from hepatotoxins. Appropriate citations should be included in this aspect.
A1: We agree that liver regeneration after hepatotoxin injury may be delayed and less reproducible compared to ALF after liver resection. Appropriate references from rodent models have been added to underscore this point (Takada, 2003; DOI: 10.1007/s100470300001)
Q2) There is a lack of histologic evidence of regeneration of the native liver in the ALF models. This is important since drug induced ALF is a leading cause of liver toxicity and the short term benefits from metabolic support from SBARS does not necessarily address the real world problem of donor shortage unless the metabolic support can be shown to continue until the time that the native liver begins to regenerate to obviate the need to OLT.
A2: All three studies report histological evidence of liver regeneration by Ki-67 staining. Tissue was available at baseline and at autopsy in all study animals. However, liver biopsies were not possible at other time points after induction of ALF due to the risk of bleeding. In addition, we report liver regeneration in terms of CT scan liver volume at 43hrs (peak response) and 90hrs in our 85% hepatectomy study. Lab values associated with liver function were also measured and indicate increased liver functional capacity in the pigs treated by SRBAL compared to control pigs. The discussion has been revised to underscore these points.
Q3) Benefits of SRBAL appears to be maximum when initiated immediately after the induction of ALF and in one of the models, survival decreased significantly when started 12 hours after ALF. This should also be discussed as a limiting factor as patients do not always present to hospitals early in the course of ALF.
A3: Yes, patients in ALF often arrive at later stages of illness which may diminish the benefit of therapy. In our studies, early vs. later initiation of therapy was a study variable in the primate study. In the other two (pig) studies, therapy was initiated at onset of ALF, i.e., coagulopathy, hepatic encephalopathy) in-order-to achieve clinical relevance and improve prediction of benefit in future clinical trials. These points are highlighted in the revised manuscript.
Reviewer 2 Report
Thank you for asking me to review this interesting manuscript in which the authors have performed a meta-analysis of three experimental models of liver failure in which their Spheroid reservoir BAL (SRBAL) has been tested. The three models used different time scales and instigators of liver failure and all models were intended to induce liver failure prior to “treatment” with the BAL although the time from instigation of damage to treatment varied from 12 to 48 hours, and periods of treatment also differed. They then analysed the results of each model and attempted to draw and pool conclusions from each. Their conclusion was that successful data indicated the SRBAL could now be tested in patients.
This article gives a good introduction to the bioartificial liver field (BAL), and the strength of data suggesting that BAL has a role to play in patients with liver failure.
The supplementary video is a pictorial video of the specific device used, that is similar to Figure 1, but easy to assimilate for lay readers.
The methodology states what each experimental model tested for, and naturally referred back to the several previous papers from the authors, in this area. The experimental models used pigs in two studies and rhesus monkeys in one study, treated at different times from the liver insult and for different times and treatment regimes.
The results have presented data from all three models in two species, and their conclusions are drawn from the pooled data.
The discussion considers some limitations of the specific models used, and has considered the potential risk of zoonosis, to some degree. It does not discuss the contribution from the albumin circuit compared with the cell component; both SRBAL and NCBAL contain the albumin circuit it would appear, from the description. If that is not the case it should be clearly stated. If that is the case then there should be a discussion as to the limitations of using hollow-fibre cartridges.
The authors discuss a use of 200grams of hepatocyte spheroids. Please state how many cells that equates to and what the viability of those spheroids was at start and end of treatment for each of the experimental model groups.
Two of the models were drug induced, each different, and one was surgically induced. However, it is well recognised that not every animal (nor indeed human) will respond to a given dose of drugs in the same way, and thus there can be bias in the data; I imagine the authors also realised this and thus used a surgical model, so that all animals would react the same way. However, the surgical model that was chosen was 85% partial hepatektomy, where the remainder of the liver was “healthy”, and it is also widely known that patients after hepatic resection for cancer recover well and quickly, so this does not represent liver failure, although 85% hepatektomy does represent a paucity of liver function, and could lead to liver failure in time however, this study started at 24h after induction and lasted 24h. It is noted that 3 pigs were followed up for 2 weeks, however, it is not clear if this represented a survival of three out of the 18 used in the study?
If looking at a meta-analysis, a Forest plot may be more representational than simply a table of p-values, and should be shown. Moreover it may be necessary to utilize an extra test such as Browns or Harmonic mean p value since it is not clear in the text whether the tests chosen are dependent or independent. Given that most are arising from liver failure, it is likely that they are dependent.
Meta-analyses are considered to be appropriate if each comparator (in this case each number of animals in each model) have n=>50. This is not the case in this study, so whilst the data presented is of interest to the field, can it really be considered a true meta-analysis? Please justify.
Figure 2 Kaplan Meier survival curve shows pooled survival data from all of the models but it does not indicate, nor could I find in the discussion, the expected survival times for each experimental model. This should be added, since the Kaplan Meier curve may be weighted by one particular set of data from one of the three experimental models. As each model was tested in different numbers of animals this could influence the outcome.
In the text alluding to Table 3, the authors state in several places that INR data is shown, indicating coagulation profile, however that data does not appear either in text or table; please add.
It was not stated whether each of the studies utilised the same anaesthetic regime/components. Given that liver failure is usually associated with raised intracranial pressure (ICP) it should be stated if they are not all using identical anaesthetics. For example Propafol, decreases ICP, and unless it was used in all studies can influence the true data.
In the text the control groups used are denoted are SMT and NCBAL, whereas in Table 2 legend and table itself SMT is shown as SMC and NCBAL as BAL. The text, tables, figures and legends should be consistent throughout.
Figure 2 Kaplan Meier survival curve has no legend. Moreover it is incorrectly labelled as Fig 2. It should be labelled Figure 3, and also when referred to in the text since Figure 2 appears earlier where the design of all studies is shown.
Table 1 and Table 3 should if possible each appear on a single page as the multiple columns are hard to follow when split into two pages, as they are in the downloaded version I see to review.
Table 1 dose of galactosamine is missing. Please add.
Author Response
Q1: Thank you for asking me to review this interesting manuscript in which the authors have performed a meta-analysis of three experimental models of liver failure in which their Spheroid reservoir BAL (SRBAL) has been tested. The three models used different time scales and instigators of liver failure and all models were intended to induce liver failure prior to “treatment” with the BAL although the time from instigation of damage to treatment varied from 12 to 48 hours, and periods of treatment also differed. They then analyzed the results of each model and attempted to draw and pool conclusions from each. Their conclusion was that successful data indicated the SRBAL could now be tested in patients.
This article gives a good introduction to the bioartificial liver field (BAL), and the strength of data suggesting that BAL has a role to play in patients with liver failure.
The supplementary video is a pictorial video of the specific device used, that is similar to Figure 1, but easy to assimilate for lay readers.
The methodology states what each experimental model tested for, and naturally referred back to the several previous papers from the authors, in this area. The experimental models used pigs in two studies and rhesus monkeys in one study, treated at different times from the liver insult and for different times and treatment regimes.
The results have presented data from all three models in two species, and their conclusions are drawn from the pooled data.
A1: Thank you very much for the positive feedback.
Q2: The discussion considers some limitations of the specific models used, and has considered the potential risk of zoonosis, to some degree. It does not discuss the contribution from the albumin circuit compared with the cell component; both SRBAL and NCBAL contain the albumin circuit it would appear, from the description. If that is not the case it should be clearly stated. If that is the case then there should be a discussion as to the limitations of using hollow-fibre cartridges.
A2: Thank you very much for the question. The albumin circuit contains 5g/L human albumin in the SRBAL and the NCBAL treatment groups. The efficiency of this treatment system has already been proven in bench studies. To identify the effect of the albumin circle in consideration of the individual cartridges, the NCBAL regime was always compared to the SMT regime. The similarity of the results in both groups excluded a treatment success of the albumin circulatory alone.
Q3: The authors discuss a use of 200grams of hepatocyte spheroids. Please state how many cells that equates to and what the viability of those spheroids was at start and end of treatment for each of the experimental model groups.
A3: Two hundred grams of hepatic spheroids are approximately equivalent to 20x109 hepatocytes. The stable variability of the hepatic spheroids was already shown in bench studies previously and is well known (Nyberg et al., 2005; DOI: 10.1002/lt.20446). However, due to the potential decrease of selective hepatocyte functions like albumin production, a replacement of the spheroid after 24 hours should be recommended. The introduction and discussion part get modified to contain this information.
Q4: Two of the models were drug induced, each different, and one was surgically induced. However, it is well recognized that not every animal (nor indeed human) will respond to a given dose of drugs in the same way, and thus there can be bias in the data; I imagine the authors also realized this and thus used a surgical model, so that all animals would react the same way. However, the surgical model that was chosen was 85% partial hepatectomy, where the remainder of the liver was “healthy”, and it is also widely known that patients after hepatic resection for cancer recover well and quickly, so this does not represent liver failure, although 85% hepatectomy does represent a paucity of liver function, and could lead to liver failure in time however, this study started at 24h after induction and lasted 24h.
A4: As outlined in the introduction and discussion part of the manuscript, the human ALF is a complex disease that can only be represented to a limited extent by a single research model. To represent the human situation as closely as possible, a combination of different ALF models in different large animal species is required. Demonstrating treatment success in both surgical and nonsurgical research models acknowledges this fact. However, clinical studies need to follow to demonstrate the effect of SRBAL treatment in humans.
Q5: It is noted that 3 pigs were followed up for 2 weeks, however, it is not clear if this represented a survival of three out of the 18 used in the study?
A5: This point has been clarified in the Result section
Q6: If looking at a meta-analysis, a Forest plot may be more representational than simply a table of p-values, and should be shown. Moreover it may be necessary to utilize an extra test such as Browns or Harmonic mean p value since it is not clear in the text whether the tests chosen are dependent or independent. Given that most are arising from liver failure, it is likely that they are dependent.
A6: For this meta-analysis, we used a log-rank test to match the results of the results of the three included studies. This test does not report the effect size and confidence intervals of the analyzed data, which would be necessary for a Forest plot. Therefore, we can only present the results in a general overall table.
A Brown’s or Harmonic mean p-values requires dependent p-values. Since the experiments were from different studies and on different animals, the p-values are independent. Due to this circumstances, the Fisher’s method is more appropriate for the data analysis.
Q7: Meta-analyses are considered to be appropriate if each comparator (in this case each number of animals in each model) have n=>50. This is not the case in this study, so whilst the data presented is of interest to the field, can it really be considered a true meta-analysis? Please justify.
A7: Sample size calculation in large animal preclinical studies depends on several factors. In addition to the power of the study, ethical concerns must be considered during the design of these types of studies. (Serdar et al, 2011; DOI: 10.11613/BM.2021.010502). Due to these circumstances, generalized statements regarding the minimum sample size for this particular type of scientific study, as well as resulting meta-analyses, are challenging to make. The calculation of each individual included study includes a separate calculation of sample sizes calculation and represents the fundament of our resulting meta-analysis. Therefore, in the context of preclinical large animal research and under consideration of multiple factors like ethical concerns, the reported group sizes are sufficient to qualify the presented analysis as a "true meta-analysis."
Q8: Figure 2 Kaplan Meier survival curve shows pooled survival data from all of the models but it does not indicate, nor could I find in the discussion, the expected survival times for each experimental model. This should be added, since the Kaplan Meier curve may be weighted by one particular set of data from one of the three experimental models. As each model was tested in different numbers of animals this could influence the outcome.
A8: Thank you very much for the question. The expected survival curves for each individual model was added to the Kaplan Meier curve.
Q9: In the text alluding to Table 3, the authors state in several places that INR data is shown, indicating coagulation profile, however that data does not appear either in text or table; please add.
A9: INR/coagulation profile was only measured in two of our studies (Glorioso et al. and Chen et al.). This information has been added/clarified in the Result section of the meta-analysis.
Q10: It was not stated whether each of the studies utilized the same anaesthetic regime/components. Given that liver failure is usually associated with raised intracranial pressure (ICP) it should be stated if they are not all using identical anaesthetics. For example Propofol, decreases ICP, and unless it was used in all studies can influence the true data.
A10: All treatments in each individual study were performed under mild sedation by continuous intravenous administration of Propofol (0.1-0.2 mg/kg/min). This information was added to the method section.
Q11: In the text the control groups used are denoted are SMT and NCBAL, whereas in Table 2 legend and table itself SMT is shown as SMC and NCBAL as BAL. The text, tables, figures and legends should be consistent throughout.
A11: The abbreviations in context of SRBAL treatment were adapted and are know consistent.
Q12: Figure 2 Kaplan Meier survival curve has no legend. Moreover it is incorrectly labelled as Fig 2. It should be labelled Figure 3, and also when referred to in the text since Figure 2 appears earlier where the design of all studies is shown.
A12: The labeling of all Figures were reviewed and a legend get added by the Kaplan Meier survival curve.
Q13: Table 1 and Table 3 should if possible each appear on a single page as the multiple columns are hard to follow when split into two pages, as they are in the downloaded version I see to review.
A13: We modified the format in the current available version of the manuscript.
Q14: Table 1 dose of galactosamine is missing. Please add.
A14: The dose of D-galactosamine was added to Table 1.
Round 2
Reviewer 2 Report
This is a response to the authors comments on my initial review.
Most questions have been addressed satisfactorily, however, these below require further clarification
Q5: It is noted that 3 pigs were followed up for 2 weeks, however, it is not clear if this represented a survival of three out of the 18 used in the study?
A5: This point has been clarified in the Result section
Glorioso et al. and Chen et al. confirmed a survival benefit of SRBAL therapy in drug 234 and surgical induced porcine ALF models. In particular, Chen et al. was able to follow up 235 3 out of the 6 SRBAL treated pigs after SRBAL treatment for up to 14 days after 85% liver 236 resection.
No it hasn’t clarified if the other 3 of the six died, or if they were euthanased at an earlier time ?
Q9: In the text alluding to Table 3, the authors state in several places that INR data is shown, indicating coagulation profile, however that data does not appear either in text or table; please add.
A9: INR/coagulation profile was only measured in two of our studies (Glorioso et al. and Chen et al.). This information has been added/clarified in the Result section of the meta-analysis.
It hasn’t been added or clarified, it has simply been removed. Did they or did they not measure INR?
If not, why not?
Author Response
Dear reviewer,
See below the answers (A5.2 and A9.2) to both question Q5 and Q9
Most questions have been addressed satisfactorily, however, these below require further clarification
Q5: It is noted that 3 pigs were followed up for 2 weeks, however, it is not clear if this represented a survival of three out of the 18 used in the study?
A5: This point has been clarified in the Result section Glorioso et al. and Chen et al. confirmed a survival benefit of SRBAL therapy in drug and surgical porcine models of ALF. In particular, Chen et al. was able to follow up 3 out of the 6 SRBAL treated pigs after SRBAL treatment for up to 14 days after 85% liver resection.
Glorioso et al. and Chen et al. confirmed a survival benefit of SRBAL therapy in drug and surgical induced porcine ALF models. In particular, Chen et al. was able to follow up 3 out of the 6 SRBAL treated pigs after SRBAL treatment for up to 14 days after 85% liver resection.
No it hasn’t clarified if the other 3 of the six died, or if they were euthanased at an earlier time?
A5.2: We have now better clarified survival of pigs in the Chen study. One pig was euthanized at 44 hours after developing increased ICP and brain death equivalence. Five of six pigs survived to the primary endpoint of 90 hours. Two pigs were euthanized after 90hrs (during the second week) when they developed symptoms consistently with small-for-size graft syndrome (tense ascites, lethargy, poor oral intake). This additional information has been added in the result section.
Q9: In the text alluding to Table 3, the authors state in several places that INR data is shown, indicating coagulation profile, however that data does not appear either in text or table; please add.
A9: INR/coagulation profile was only measured in two of our studies (Glorioso et al. and Chen et al.). This information has been added/clarified in the Result section of the meta-analysis.
It hasn’t been added or clarified, it has simply been removed. Did they or did they not measure INR?
If not, why not?
A9.2 Li et al did not include the INR measurements in their study design. Consequently, we are unable to include this missing data in our meta-analysis. To avoid any bias in the comparison of the three treatment options, we removed the INR/coagulation profile from the manuscript.
Furthermore, we modified the discussion by adding a statement that two of three studies reported INR data and these two studies demonstrated evidence of INR improvement in the SRBAL treatment groups.